# Recent Advances in Non-Invasive Blood Pressure Monitoring and Prediction Using a Machine Learning Approach

**DOI:** 10.3390/s22166195

**Published:** 2022-08-18

**Authors:** Siti Nor Ashikin Ismail, Nazrul Anuar Nayan, Rosmina Jaafar, Zazilah May

**Affiliations:** 1Department of Electrical, Electronic and Systems Engineering, Universiti Kebangsaan Malaysia, UKM Bangi 43600, Selangor, Malaysia; 2Institute Islam Hadhari, Universiti Kebangsaan Malaysia, UKM Bangi 43600, Selangor, Malaysia; 3Electrical and Electronic Engineering Department, Universiti Teknologi Petronas, Seri Iskandar 32610, Perak, Malaysia

**Keywords:** non-invasive, blood pressure, continuous monitoring, sensors, machine learning

## Abstract

Blood pressure (BP) monitoring can be performed either invasively via arterial catheterization or non-invasively through a cuff sphygmomanometer. However, for conscious individuals, traditional cuff-based BP monitoring devices are often uncomfortable, intermittent, and impractical for frequent measurements. Continuous and non-invasive BP (NIBP) monitoring is currently gaining attention in the human health monitoring area due to its promising potentials in assessing the health status of an individual, enabled by machine learning (ML), for various purposes such as early prediction of disease and intervention treatment. This review presents the development of a non-invasive BP measuring tool called sphygmomanometer in brief, summarizes state-of-the-art NIBP sensors, and identifies extended works on continuous NIBP monitoring using commercial devices. Moreover, the NIBP predictive techniques including pulse arrival time, pulse transit time, pulse wave velocity, and ML are elaborated on the basis of bio-signals acquisition from these sensors. Additionally, the different BP values (systolic BP, diastolic BP, mean arterial pressure) of the various ML models adopted in several reported studies are compared in terms of the international validation standards developed by the Advancement of Medical Instrumentation (AAMI) and the British Hypertension Society (BHS) for clinically-approved BP monitors. Finally, several challenges and possible solutions for the implementation and realization of continuous NIBP technology are addressed.

## 1. Introduction

Outstanding technological capabilities in the present day have contributed significant impacts to the healthcare sector via continuous monitoring of vital signs such as blood pressure (BP) for home users. These health monitoring tools, such as cuff sphygmomanometers for measuring BP non-invasively, are extensively used by patients with various health conditions, such as cardiovascular-related diseases and diabetes, or even by subjects with no history of ailments.

BP is a measurement of force exerted by the heart against the arteries during the pumping of blood and it is used by health practitioners to evaluate a patient’s current state of health. To date, there are two clinical, gold-standard ways of monitoring BP. The first one is performed invasively in intensive care units (ICUs) or operative ward environments via arterial catheterization, while the second approach is more flexible because it measures BP non-invasively and so it is more suitable for clinical or at-home applications using the cuff sphygmomanometers. In comparison to the invasive way, the non-invasive blood pressure (NIBP) measurement method shows superior performance in terms of ease of use, low cost, portability, reproducibility, and simplicity in that it does not require the judgement of experts to understand the estimated BP.

For an individual with non-existing conditions, the ideal systolic blood pressure (SBP) and diastolic blood pressure (DBP) according to the American Heart Association are less than 120 and 80 mmHg, respectively [1]. Hypertension is diagnosed when the estimated SBP and DBP values are beyond the considered range, whereas hypotension is diagnosed when these values are below this range. The World Health Organization (WHO) reported that about half a billion hypertension patients were never diagnosed despite suffering from cardiovascular diseases (CVD), and 720 million patients were left untreated from 1990 to 2019 [2]. Although patients undergo frequent medical check-ups, this silent killer disease is difficult to detect, which results in delayed prognosis and fatality [3]. Several risk factors of this illness are attributed by hypertension, diabetes, and lipid anomaly [4,5,6,7,8,9].

The first experimental realization of non-invasive BP monitoring, by von Basch in 1876 [10], used a mercury manometer to measure BP and eliminated artery puncture. Two decades later, Riva-Rocci reported the first clinically-accepted sphygmomanometer, which relies on artery compression by a rubber cuff in a circular direction to measure arm pressure with improved accuracy and less pain [11,12]. This method produces erroneous results when used to measure SBP because of the narrow cuff width of 5 cm [13], which was thus later increased to 12 cm [14]. In 1981, Donall Nunn invented the first fully automated oscillometric cuff BP monitor with high estimation accuracy [15].

Currently, the cuff-based technique possesses few limitations that are still considered as ongoing roadblocks in clinical medicine for finding suitable alternatives to measure BP unobtrusively and targeted for general populations. This is because patients often experience pain and discomfort [11] during BP recording because of the cuff compression, and this raises the issue of poor cuff fitting on individuals with obesity [16] and neonates [17]. Additionally, the tendency of failure detection for hypertensive patients is high because the associated participants portray normal BP values during measurement, suggesting the masking of hypertension [18]. This phenomenon is in contrast to the white-coat effect in which normotensive subjects are misidentified as hypertensive partly because of anxiety prior to the measurement session or post-exercise effects [19]. Moreover, this approach measures BP values intermittently, which does not adequately provide detailed real-time information on an individual’s state of health. A study found that great BP variation with valuable information related to CVD detection can be observed during sleep [20]. In addition, this modality requires a stationary position of the participant during measurement to avoid motion artifacts introduced by noises, hence producing unwanted signals that interfere with pulse recording [21]. Other factors, such as the banning of mercury sphygmomanometers in United States hospitals due to mercury toxicity in the environment [22], variation of BP values across individuals because of artery occlusion [23], severe rupture of skin capillary as a secondary effect of exposure to cuff BP [24], and faint or almost inaudible Korotkoff sound between systolic and diastolic recording when the cuff deflates for cardiac arrythmias patients [25], have led to the investigations of a variety of solutions in recent years to overcome the aforementioned drawbacks of cuff BP monitoring devices.

To the best of our knowledge, the cuffless NIBP sensors are yet to be validated in a way that can be readily implemented as the gold standard of the NIBP monitoring model. The alternatives have received considerable critical attention as they are capable of estimating continuous and accurate BP results with the aid of advanced technology such as machine learning (ML), and these can be expanded for remote and ambulatory (24 h) monitoring [26,27] through microcontrollers and gateways with the goal of improving healthcare quality by helping patients to receive optimal treatment in a timely manner and personalized health monitoring for everyone.

This review summarizes available state-of-the-art NIBP sensors, as displayed in Figure 1, and identifies several commercialized NIBP monitoring devices for continuous BP measurements studies. Furthermore, the prediction techniques of continuous BP monitoring, including pulse arrival time (PAT), pulse transit time (PTT), pulse wave velocity (PWV), and ML, are elaborated based on acquired bio-signals from these NIBP sensors. On the basis of several reported studies, the estimated values of SBP, DBP, and mean arterial pressure (MAP) from various ML models are compared according to universal validation protocols for clinically-approved BP monitors. Finally, the challenges in this field and possible improvements for future works are addressed in this paper to ensure the implementation and acceptance of this emerging technology in today’s world.

## 2. Sensors for NIBP Monitoring

NIBP sensors offer continuous monitoring of BP, which is an important biomarker in assessing individual state of health in real time. In this section, the various NIBP sensors based on optical, electrical, pressure, and ultrasound principles are reviewed.

### 2.1. Photoplethysmogram

Glenn Allan Milikan devised the first oximeter in 1940 and introduced the term “oximeter” a year later [28]. In 1973, the Czesch physiologist Jan Peňáz introduced the volume clamp technique, where arterial pulse obtained from a finger is indirectly measured through vascular unloading [29]. In the 1980s, photoplethysmogram (PPG) proliferation in clinical applications was observed through the discovery of pulse oximeters for continuous oxygen saturation (SpO_2_) monitoring under anesthesia [30]. Recently, there has been widespread usage of PPG as pulse oximeters, especially by COVID-19 patients throughout the home surveillance period. With the current breakthrough of PPG-based BP measurement frameworks, continuous monitoring using cuffless sensors seems possible as several advantages are associated with this sensor including its portability, reliability, easily operated, and cost-effective [31,32].

The generation of a PPG pulse is based on an optical principle that involves the interaction between a light-emitting diode (LED) and a photodetector (PD). LED emits light on peripheral tissues, and then the light is scattered to the extent where only a proportion of the light is absorbed by the tissues. PD measures differences in absorbed light as arterial blood volume changes during a cardiac cycle [33]. Typically, a PPG waveform consists of three significant points: systolic peak, dicrotic notch, and diastolic peak, as shown in Figure 2. These peaks define the highest and lowest BP values during one cycle of heart activity; meanwhile, the notch represents the backflow of blood into the heart, which causes the closure of the aortic valve [34].

For PPG-based BP prediction using PTT, two PPG sensors are placed further from each other either at the same location or multiple peripheral sites. These sites could be from the temporal artery (forehead), carotid artery (ear and neck), brachial artery (arm), radial artery (wrist and finger), posterior tibial artery (ankle), and dorsalis pedis artery (foot) [35]. However, there is a lack of evidence for the best selection of BP measurement sites provided so far [36].

Finapres (Ohmeda, Model 2300) is an example of a commercialized digital PPG finger cuff with a pre-installed infrared (IR) photocell, a manometer, an automated inflated cuff, and a BP monitor. The utilization of Finapres as an arterial BP estimator was based on a volume clamp to measure a continuous beat-to-beat finger BP waveform and generate SBP, DBP, and MAP digitally. A model of continuous BP prediction was developed in [37] for 27 healthy individuals based on photoplethysmogram intensity ratio (PIR) and PTT. The estimated BP from that proposed method were compared with Finapres with an accuracy of −0.37 ± 5.21, −0.08 ± 4.06 and −0.18 ± 4.13 mmHg, for SBP, DBP, and MAP, respectively. It was found that the PIR-PTT method displayed higher correlation with Finapres than those achieved with PTT algorithms. Further, Portapres is another readily available PPG-based device that produces continuous results. Another work investigated the relationship of depression and nocturnal BP of 19 adolescents during sleep and wake states through continuous BP monitoring [38]. It was found that the Portapres model-2 (TNO-TPD, Biomedical Instrumentation, Amsterdam, Netherlands) was able to measure BP values with very low disturbance, and greater SBP recordings were observed for depressed subjects with an average of 11 mmHg difference from healthy groups. Nevertheless, more efforts are required to extend both works on larger sample size for more comprehensive evaluation of predicted BP.

PPG sensors display several limitations that require extensive efforts to improve the efficacy and feasibility of PPG as an alternative for NIBP monitoring. One work on BP estimation based on dual PPG sensors observed a dramatic reduction of PWV variations only after the applied pressure at the measurement site was kept constant. This shows that the susceptibility of the PPG signals to applied pressure might contribute to substantial effects, such as artificial alteration of the PPG waveform [39]. Loss of dicrotic notch has also been observed in the PPG waveform of the elderly because of aging [40], which is an independent factor of CVD development [41]. The raw PPG signals are often contaminated with various types of noises and require signal processing techniques to isolate the clean PPG signals from uncorrelated noises for further prediction analyses, as depicted in Figure 3.

### 2.2. Electrocardiogram

Electrocardiogram (ECG) has been employed by clinicians to measure cardiac physiological signals non-intrusively, including heart rate (HR) estimation, heartbeat rhythm regularity assessment, and CVD diagnosis. Even though the ECG waveform only displays the heart’s electrical activity and reflects no BP information [42], the integration of ECG and PPG sensors in the same system has been extensively studied for continuous BP monitoring. This is because a high BP prediction accuracy will be generated and the approach can be adopted in wearable devices, despite both signals depicting different working principles.

The ECG waveform consists of electrical cardiac depolarization and repolarization phase recordings measured in the time domain [43], and the sinusoidal pattern of PQRST complexes highly correlates with heartbeat rate [44]. Based on Figure 4, the P wave indicates atrial depolarization within 0.11 s and a maximum amplitude of 3 mm. The QRS complex denotes ventricle depolarization with quick response, i.e., less than the atrial contraction duration in the P wave, and this complex has attracted attention as a CVD parameter, followed by the T wave, which portrays ventricle repolarization [45].

There are many readily available cardiograph monitors in the market, but only a few of them incorporate ECG and PPG in the same device, such as CardioQVARK and MAX86150 (Maxim Integrated, San Jose, CA, USA). CardioQVARK was developed by Russian engineers to monitor real-time cardiovascular parameters for at-home patients. For example, BP values of 500 participants in [46] were found to be highly correlated between CardioQVARK and the cuff BP monitor with an accuracy of 0.32 ± 3.63 mmHg and 0.35 ± 2.95 mmHg for SBP and DBP, respectively. Another work performed BP estimation on 140 volunteers using MAX86150 module and achieved a 5.7 ± 5.5 mmHg difference from the cuff sphygmomanometer [47]. However, this study applied extensive feature engineering work to obtain a good signal quality prior to regression analyses. A graphical representation of the raw and normalized ECG signals is shown in Figure 5.

Artifacts contamination is a key challenge for the implementation of ECG signals in BP monitoring field, especially because this sensor is sensitive to non-constrained environments that are filled with potential sources of noises, including motion artifact, incorrect electrode placement, and baseline wander. Hence, several methods have been suggested to address this issue, such as using ML models and advanced signal processing algorithms.

### 2.3. Tonometer

Arterial tonometry (AT), based on the applanation tonometry principle, is an established method of measuring arterial BP non-invasively, in which pressure is given toward the arterial line causing the compression of the arteries, thus producing arterial pressure waveforms. BP can be estimated through direct and indirect techniques of AT [48].

For the direct method, BP values are estimated from the recording of pressure waveforms at the common carotid artery. The reason behind this process is that the waveform shape of the common carotid artery resembles that of the aorta, giving an advantage to the prior method in terms of accessibility [49]. Thus, that the common carotid artery shows close vicinity to the aortic BP measurement site in AT. Indirect estimation begins with pressing a hand-held tonometer on the radial artery with low pressure to generate radial artery pressure [50]. The latter pressure is detected by a sensor and forms a radial artery waveform. The waveform is recorded and initiates the formation of a central pressure waveform via the generalized transfer function [48]. Radial AT is suitable for clinical use because of its ease of measurement, better comfort, and reproducible results [51]. Furthermore, AT is a good approach for investigating left ventricular heart function for a detailed assessment [52].

BP monitors in the market that operate with AT setting include PulsePen (DiaTecne, Milan, Italy) [53], SphygmoCor XCEL (AtCor Medical, Sydney, NSW, Australia) [54], and Omron HEM-9000AI (Omron Healthcare, Kyoto, Japan) [55]. A recent work evaluated the prediction accuracy of central BP for 20 subjects during rest and exercise stages using the SphygmoCor XCEL and a cuff sphygmomanometer [56]. The results show that BP values between both techniques are comparable during steady-state and low intensity exercise with a difference of 0.6 ± 5.4 mmHg and 0.8 ± 2.1 mmHg for SBP and DBP, respectively. Further, the cuff BP monitor lacks the capability to measure BP during a high intensity state due to the exertion caused by movement, which leads to pulse signals distortion. Another study compared BP values measured using the BPro^®^ tonometer sensor (Healthstats, Singapore) and a cuff sphygmomanometer to determine which modality best predicted chronic kidney disease (CKD) obtained from 10,197 subjects [57]. It was found that both techniques offer promising results as CKD predictors, but the SBP and DBP estimated by the cuff BP monitor are more biased towards women than men, with lower BP values. This indicates that with BPro^®^ employment, it may be a great surrogate to the current (goal) standard BP monitoring tool and more efforts can be done to study its usability as a CKD risk predictor especially in women, as this group experiences the inevitable process of menopause, which may contribute to stiffer aorta.

However, the pulse waveforms obtained from AT contain limited information, such as pulse pressure recordings, and this method is incapable of providing absolute SBP and DBP values [19], systemic errors caused by cuff-based calibration [58], and poor prediction accuracy when calibrated with invasive catheterization [59]. Recently, the risk stratification of employing AT for estimating BP in the carotid artery has increased; consequently, this method shows declining utility trends in clinical settings because of the difficulty of achieving sufficient applanation, the possibility of thrombus displacement [60], and the calibration inaccuracy of a brachial sphygmomanometer. The author suggested that carotid BP AT should be applied with mild pressure and regular checking of central pressure waveforms should be avoided.

### 2.4. Ultrasound

The prediction of BP using an ultrasound (US) approach is not supported by theoretical principle; nevertheless, algorithms such as PWV and Moens–Korteweg (MK) equations are significant for enabling the indirect estimation of BP values as well as solving computational related problems [61]. US is based on the concept of sound waves that propagate in a medium with pulse frequencies beyond the limit of human being’s hearing ability [62]. It can be used to measure blood velocity, blood flow, and the imaging of an artery embedded in organs [63] such as the heart and uterus. US probe or transducer plays a vital role in capturing and displaying signals. When electricity flows through a transducer, it transmits wave pulses (signals) and hits the subject of interest, such as blood cells. Then, the reflected signal from blood approaches the transducer before it generates signal waveforms as output [64].

A previous study [65] reported that US Doppler is a better marker than the cuff sphygmomanometer for blood flow speed monitoring through the probe placement over the brachial artery for patients diagnosed with muscular atrophy, in which the Korotkoff sounds are almost inaudible [25]. The main factors contributing to the widespread application of the Doppler concept in US for measuring BP are its ability to penetrate deep tissue and vessels [66], to monitor dimensions of vessel wall in longitudinal and transverse orientations [67], and to calculate blood flow velocity [68].

There are various groups currently working on US-based BP detection studies through different approaches. Meusel et al. [61] compared the commercialized US transducers with other NIBP monitoring methods to validate the potential application of US in continuous BP measurement of 10 healthy subjects. The US transducers models used in this work are Siemens Acuson S2000, Transducer 14L5 (Siemens Healthcare, Erlangen, Germany) and Philips iU22, Transducer L17-5 (Philips Medical Systems, Hamburg, Germany). Both devices are then validated with reference BP from two sources: the cuff sphygmomanometer and the CNAP™ monitor (Monitor Dräger Infinity Delta, Luebeck, Germany), where all of these are placed at three measurement sites and greater prediction accuracy is depicted by US monitoring tools than continuous non-invasive arterial pressure (CNAP) when compared to cuff BP.

In a different approach, Zakrzewski et al. [69] measured arterial tissue displacements using US strain elastography and converted these to estimate pulse pressure using algorithms. Slow compression sweeps were applied to the carotid artery and the results demonstrated the pressure waveforms, as shown in Figure 6a. It was found that the artery was greatly compressed as an effect of increasing the applied force, suggesting the resistance of the artery wall against deformation [70]. In addition, the peaks and throughs of the pressure waveform are labelled as SBP (red) and DBP (blue), respectively. This work is an extension of previous study [71] that included real subjects with no history of CVD and hypertension over the course of one month. Figure 6b displays BP sweeps of a hypertensive subject. It was observed that since the first sweep, the BP level kept decreasing until it reached to a point slightly before the tenth day, where it started to show a noticeable increase. The reported results could be due to the history of hypertension, which confirmed the prevention of any BP spike during the earlier sweeps through medication intake. Additionally, the predicted SBP and DBP were comparable with the cuff-based BP values. Although the utilized approach of continuous BP estimation varies from one group to another, the proposed works achieved satisfactory results, and this could be a groundwork for bridging the gap in continuous NIBP monitoring enabled by US.

Several weaknesses of the US method for NIBP monitoring are cumbersome US probe placement procedures, the need to have skilled operators, sensitivity to motion, and bad contact between the US probe and the skin area [72]. This is because of the uneven layer of applied US gel on the skin surface, which will attenuate the US wave and reduce the prediction accuracy. Additionally, a patient was reported to have an allergen reaction towards the US gel’s composition, which resulted in contact dermatitis [73].

Overall, regardless of the sensing mechanisms, these NIBP sensors offer promising potentials for guiding future research of continuous BP prediction. For example, the widespread usage of pulse oximeters in a clinical environment indicates the feasibility of optical-based sensors capturing a human bio-signal, even though the BP estimation using PPG is still in its infancy stage. In addition, the integration of PPG and ECG in the same platform provides valuable discoveries including high BP signals quality, thus showing the importance of ongoing efforts to improve the reliability of this system. Conversely, low research works are reported on the AT-based BP prediction approach due to several factors, including poor DBP estimation accuracy and sensor errors. From the aspect of US BP monitors, increasing attention is being paid to improve device portability, low cost and easy operation since the current ones in the market are not suitable for continuous BP monitoring.

## 3. Continuous BP Prediction Techniques and Validation

Having investigated the different sensing mechanisms of NIBP sensors, this section discusses the prediction approaches of continuous BP monitoring based on acquired signals from NIBP sensors and compares the SBP, DBP, and MAP results estimated using various ML algorithms in several reported works based on the international protocols for clinical verification of BP monitoring devices.

### 3.1. Pulse Arrival Time (PAT)

One way of predicting BP values continuously is by performing the PAT calculation on acquired pulse signals from PPG and ECG type of sensors. PAT can be expressed as the time taken for the pulse wave to travel from the heart to the peripheral location. In general, the PAT at the specified points of the PPG signals was derived from the beginning of the heartbeat at the R-peak of the ECG, as displayed in Figure 7.

The points could be from the R-peak of the ECG signal to the foot of PPG (${\mathrm{PAT}}_{f}$), the mid-point of the PPG (${\mathrm{PAT}}_{d}$) or the peak of the PPG (${\mathrm{PAT}}_{p}$) waveform. The measurement of PAT usually includes the pre-ejection period (PEP) that incorporates the heart’s electromechanical delay, which explains the relationship of the PEP and PTT, as shown in Equation (1).
(1)PAT=PEP+PTT

Bote et al. [74] developed various PAT-based models for the continuous BP prediction of 53 patients from the Medical Information Mart for Intensive Care (MIMIC) database and performed frequent calibration with the cuff BP monitor. However, the result was that only DBP estimation achieved a passing grade according to the AAMI standard, suggesting further efforts are needed to understand the effect of PEP to PAT. Yoon et al. [75] investigated the relationship between pulse wave analysis (PWA) and BP on a sole measurement site with features extracted from ECG and PPG for PAT derivation. It was found that the proposed method is challenging due to the motion artifact as the confounding factor, yet, the PWA-BP model is feasible for the continuous cuffless NIBP measurement through PAT-derived information. Tang et al. [76] developed a predictive-based system for continuous NIBP monitoring using PPG and ECG mounted on an armchair through PAT calculation. They applied this system to healthy volunteers only and obtained low DBP values during measurements.

Wong et al. [77] studied the ability of PEP in estimating BP from 22 healthy volunteers and compared the results with the cuff monitor as the reference. The PAT technique predicted SBP with a 0.0 ± 6.6 mmHg difference from the reference BP. This work proposed the capability of PEP as a surrogate of the PTT-based BP prediction. PEP can be used to convert the electrical activity of the heart to the pumping of the mechanical heart, which can be abruptly altered because of psychological behavior and aging. Despite that, there are few disagreements of PEP inclusion in BP-related works, as discussed in [78]. The PAT–BP relationship has a few drawbacks, such as less significant PAT features than PPG features as estimated from online PPG and ECG signals [79], lack of calibration trials, and asynchronous signals of HR and BP.

### 3.2. Pulse Transit Time (PTT)

Another underlying concept of estimating BP continuously is through the PTT calculation, where it is measured from the time interval between the pressure generation at heart caused by ventricular depolarization and blood volume increase at a peripheral location due to pumping of blood out of the heart [80]. In short, PTT can be defined as the time delay between two measurement points along the arterial tree using pulse signals generated from PPG-ECG or dual PPG sensors. From Equation (1), PTT can be calculated by subtracting the PET from the PAP. The hypothesis of BP–PTT relationship is shown in Equation (2):(2)$$\mathrm{BP}=\frac{A}{{\mathrm{PTT}}^{2}}+B$$
where A and B are constant parameters that varied depending on individuals [81].

Figure 8 depicts the comparison between PEP and PTT calculations based on the ECG and the PPG signals. Each parameter such as PEP, PAT, and PTT can be elucidated as the following. As mentioned before, the time delay between the QRS complex and the beginning of the ventricular ejection of the ECG signal is measured as PEP. Further, PAT is calculated as the time delay between the R-peak of the ECG signal and the foot of the PPG signal. For the case of PTT-based calculation from dual PPG sensors, the measured time delay begins from the foot of the first PPG signal known as the proximal point and ends at the distal point where the foot of the second PPG signal is located.

Block et al. [82] compared the PTT waveforms measured at six different measurement locations on 32 subjects using ECG and PPG sensors as well as validated each BP with the cuff sphygmomanometer. The PTT-based measurements predicted BP with the least correlation when compared with the cuff BP due to the poor detection of the PPG signals. Another work by Miao et al. [83] estimated BP using a fusion-based model from a multi-sensor platform that consists of ECG and other types of sensors on 85 volunteers. The proposed method yielded better prediction accuracy when calibrated with cuff BP frequently. In addition, it was observed that the inclusion of PEP in PTT calculation contributed to consistent BP readings, which could be potentially influenced by the further propagation of the pulse wave in the body. Nonetheless, the PTT approach is still insufficient to validate the best prediction method that will portray high accuracy, low error deviations, and robustness against a dynamic environment, as the human cardiovascular system is complex and challenging for a beat-to-beat measurement.

### 3.3. Pulse Wave Velocity (PWV)

Notably, several papers reported on the counterpart of PPT, which uses a PWV calculation to estimate BP values. According to the MK equation, PWV can be defined as the velocity of the pulse wave travelling along the artery vessel and similarly, the correlation between PWV and BP can be defined as shown in Equation (3):(3)$$\mathrm{PWV}\;=\frac{L}{\mathrm{PTT}}=\sqrt{\frac{\mathrm{hE}}{\rho d}}$$
where PTT is the time taken for a pulse wave to reach from a proximal point to a distal point in a specified length of blood vessel, denoted as L [70,71]. h is the artery wall thickness, E is Young’s modulus of the arterial wall’s elasticity, ρ is the density of blood, and *d* is the distance [84].

Li et al. [85] compared the estimated BP measured using a mercury sphygmomanometer, an electronic cuff BP monitor, and the proposed method that comprises a pulse sensor and ECG. The reported findings are as follows: the PWV-based BP prediction model passed the AAMI standard, the accuracy improved by 58%, and there were mean differences of 3 ± 2.5 mmHg and 4 ± 3 mmHg for SBP and DBP, respectively, when compared with both sphygmomanometers. Byfield et al. [39] estimated the PWV-based BP estimation of 26 volunteers measured on the same finger at different locations, fingertip and below finger, using dual PPG sensors. It was observed that PWV variations reduced dramatically after the application of constant pressure at the contact area, and the predicted BP achieved mean errors of 2.117 ± 0.257 mmHg and 2.935 ± 0.721 mmHg for SBP and DBP, respectively.

Based on the above-mentioned techniques, it can be deduced that they are PPG-dependent, although several studies explored the sole utilization of ECG signal. The computational complexities of such studies are often higher and require knowledgeable experts. Nevertheless, there is no clear evidence that BP is best estimated by PAT, PTT or PWV. Hence, more research is required to understand these methods and evaluate the accuracy according to the international validation standards.

### 3.4. Machine Learning (ML)

ML is the ability of self-learning from past experiences to generate desired results, such as prediction, classification, or feedback [86]. This technique has shown tremendous potentials in disease prediction [87,88,89,90]. BP estimation obtained from NIBP sensors, such as PPG and ECG, requires signal analysis, including pre-processing, feature extraction, statistical analysis, and regression or classification. Figure 9 displays the flowchart of signal analysis performance to estimate BP for continuous NIBP monitoring.

Once the signal acquisition process completes, pre-processing is usually performed on the signals prior to prediction analysis to minimize noises by filtering [91]. Figure 9 represents the dual PPG signals after normalization using a Butterworth bandpass filter [39]. Feature extraction is an act of building related features from designated information that aims to enhance the learning performance in order to provide better prediction [92]. Signal decomposition helps produce important information and decomposes the signal into scales; this technique, such as wavelet transform, has been widely used in frequency-domain investigations [93]. Statistical analysis is subsequently conducted to identify the significant discrepancies between the predicted and reference variables. Several methods associated with this process are paired the *t*-test [94] and the coefficient of determination [95]. The signal quality index (SQI) is an important metric for assessing signal quality [96]. The final stage of BP monitoring studies involves ML regressors to perform estimation based on SBP, DBP, and sometimes MAP values, where the results are denoted as mean absolute error (MAE) and standard deviation (SD) or MAE ± SD [97].

BP prediction based on ML approaches covers the estimation of MAP, which is significant aside from SBP and DBP values for validation of the proposed models to estimate BP. MAP is defined as the average arterial pressure measured in a complete cardiac cycle, which consists of systole and diastole phases [98], as displayed in Equation (4) [99].
(4)$$\mathrm{MAP}=\frac{\mathrm{SBP}+(2\;\times \;\mathrm{DBP})}{3}$$

The most frequently used algorithms in ML for BP prediction are random forest (RF) and k-nearest neighbors (KNN), among others. For instance, RF is made up of multiple decision trees and provides prediction based on the average votes from the trees. Meanwhile, KNN learns from storing information and provides output based on comparative resemblance of datasets using the closest neighbours concept [100]. Several works on ML-based approaches for BP prediction based on the bio-signals obtained from NIBP sensors are reviewed as following.

Wu et al. [101] attempted to estimate BP by using a bio-signal device developed with an embedded ECG sensor and compared its results with those obtained using a mercury sphygmomanometer. BP prediction was performed using implemented neural network (NN) algorithms and ECG signals and ECG-driven information correlated with PPG. The proposed method portrayed great potentials, such as suitability for long-term NIBP monitoring, reduced pain due to the cuff, and unnecessary separate sensors for measurement. Khalid et al. [102] estimated BP values from PPG-based features via supervised random tree (RT), multiple linear regression (MLR), and support vector machine (SVM) algorithms for different states of BP. The approach used in this study is different from the classical PTT approach in that it extracts features from a single site of PPG instead of two distal points of sensor placement, usually PPG and ECG. However, RT performs the best in the normotensive category rather than in the hypertensive category, despite achieving the highest prediction accuracy among other models owing to small normotensive datasets.

Mejía-Mejía et al. [103] extracted the pulse rate variability features of PPG and PPG derivative (first and second) signals obtained from the MIMIC II database for classification tasks of BP. Results show that KNN outperforms other models with a maximum accuracy of 83.08 ± 1.48% for identifying BP in hypertension state. Conversely, El-Hajj et al. [104] reported that MLR depicts the lowest performance among other models and found a non-collinearity between features of PPG and BP, suggesting that ML alone is insufficient to predict BP accurately and continuously. Therefore, deep learning (DL) approaches have been incorporated to predict BP from the MIMIC II database. As a result, the estimated SBP and DBP values improve greatly with estimation errors reduced by three times than those of MLR.

DL is an extension of ML. It is a powerful method that enables complicated automated feature extraction with high performance in big data, and it reveals implicit information and automated learning without hand-crafted feature selection because of neuron layer stacking, better known as deep NN [105]. DL is also reportedly employed in image processing in which it can be used to visualize the arterial BP waveform. Qin et al. [106] proposed a novel method to visualize the BP waveform from a single PPG signal with the adoption of a regularized deep autoencoder. This study obtains a satisfactory BP waveform, and further optimization can be performed on the developed method for wearable BP sensor integration.

For BP estimation works based on bio-signal processing, convolutional neural network (CNN) and long short-term memory (LSTM) networks are among of the DL models that are frequently adopted by the biomedical engineers and researchers. Several investigations on BP prediction using these models are surveyed as follows. CNN is frequently utilized to learn trends of signals and to identify significant features within the signals [107], in such a way that it can be trained to detect abnormalities within ECG signals [108]. Baek et al. [109] proposed a novel cuffless BP prediction method based on a deep CNN using raw signals for training without PWV feature extraction and predicted BP accurately without calibration. Ibtehaz et al. [110] proposed a novel method of mapping the PPG signals from the MIMIC II database to estimate the arterial BP waveform based on deep CNN and multi-resolution analyses. The proposed method reduces phase lag for determining the relationship of PPG with BP and validates the accuracy of developed algorithms with the employment of stand-alone sensors.

Meanwhile, LSTM network can memorize previous learnings and generally utilize sequential data [111]. It can be used in studies with ECG and BP-related problems [112]. CNN combined with LSTM performs better than single CNN and LSTM models because of its more accurate prediction based on learnings from sequential data in forward and reverse modes. For instance, Baker et al. [113] devised a hybrid network composed of CNN and LSTM to predict SBP, DBP, and MAP values obtained from raw PPG and ECG signals of the MIMIC database. All BP values fulfill the passing criteria of the Advancement of Medical Instrumentation (AAMI) standard and achieve grade A for the BHS protocol, showing satisfactory results when validated with BP values measured with a cuff sphygmomanometer. The hybridized algorithms produce estimated values of 4.41 ± 6.11, 2.91 ± 4.23, and 2.77 ± 3.88 for SBP, DBP, and MAP, respectively. Corresponding to the outstanding outcomes, the proposed scheme has portrayed good reliability, as BP predictive algorithms have the potential to be expanded into further clinical usage for realization of real-time and continuous NIBP.

### 3.5. Validation of Predicted BP According to International Standards

#### 3.5.1. Advancement of Medical Instrumentation (AAMI) Standard

The reported learnings obtained from several studies were compared to identify which method satisfies the passing criteria for AAMI and portrays high prediction accuracy at par to the reference of NIBP measurement with a cuff sphygmomanometer. The proposed NIBP device is considered as AAMI approved when respective performance metrics display less than or equal to 5 mmHg for MAE and 8 mmHg for SD with at least 85 subjects [114]. Table 1 summarizes the validations of algorithms according to AAMI. All the learnings were not calibrated prior to prediction analyses, however, calibrated BPs based on previous works [109,115] were included in the table to observe the grade changes upon calibration performance. In [103], all BP values fulfill the standard of AAMI because more features can be extracted from pulse rate variabilities (PRV) during the longer duration of the segmentation, allowing a greater BP prediction quality. In [109], it was observed that prior to calibration, the predicted SBP is unable to pass the AAMI level due to the incorrect estimation according to BP states, such as overestimation for low pressure and vice versa. In [115], it was observed that the SBP prediction based on the proposed method failed to satisfy the AAMI criteria. This is due to the target variance of SBP being doubled to that of DBP, which led to higher prediction deviations. 

#### 3.5.2. British Hypertension Society (BHS) Protocol

Table 2 summarizes the reported learnings utilized in several studies for the validation of algorithms in accordance with the BHS protocol for NIBP monitoring with the achievement of prediction accuracy as a cuff sphygmomanometer, which is regarded as the gold standard technique in this area. These learnings were not calibrated prior to the estimation of BP, however, noticeable grade changes were observed after calibration performance as portrayed in [109]. The bold word denotes the study [113] that reported the best performing learnings to predict BP, among others, in which all BP values achieved an “A” grade. According to the BHS protocol [116], three grades are assigned to a newly developed yet commercialized BP monitor. The grading depends on these elements of absolute difference values: greater than or equal to 5 mmHg, 10 mmHg, and 15 mmHg. The computation of absolute difference for each category is in percentage form, and a grade of A, B, or C is designated depending on the percentage of the proposed device for every class. The eligibility of medical devices to be considered in clinical settings is exclusively for those with a grade of A or B in accordance with the BHS protocol. 

## 4. Challenges and Future Recommendations

Continual research on the development of NIBP technology faces the following key challenges for replacing the current physical measurement protocol: source of data, motion artifacts of PPG, calibration performance, dataset size, learning selection, sphygmomanometer variation models, and hampering of wires. Therefore, further efforts are needed to address these challenges and improve the efficiency of NIBP technology for continuous BP monitoring.

To begin with, online open-source databases have been widely employed in continuous BP prediction, but more offline data are required. For example, the inclusion of healthy subjects in the investigations are needed to resolve this challenge. Although MIMIC data collection requires minimal effort, the evaluation of BP accuracy is still scarce. This is because the data extracted from MIMIC contains the records of critically ill patients obtained from the ICU. Moreover, there is a loss of some of the physiological parameters during offline signal analysis, which may be caused by the ingestion of drugs and medication by patients. Hence, by selecting human volunteers as the mean of extracting BP-related features, BP variation and discrepancies can be reduced to make a good prediction.

The next challenge is the contamination of motion artifacts in the majority of NIBP sensors. Most of the data acquisition should be performed stationarily without any disturbance to avoid any distorted signal or loss of significant parameters. To address this issue, the proposed method can be trained using datasets with various kinds of interferences. As the learnings capture and store the information, they can be further tested in dynamic environments with no applied constraints to evaluate the robustness and flexibility of these models in providing great estimation. Additionally, signal processing analysis can be performed on the raw signals to smoothen the signals and remove any implicit noises and baseline wander.

Furthermore, there is a lack of an established calibration system for continuous BP monitoring. Some studies unfairly justified their novel approaches as high satisfactory without performing the proper calibration with the reference measurement of NIBP, but instead validated with other NIBP sensors such as volume clamp-based PPG, AT or an unmentioned reference source. This step should be rectified by establishing a calibration system and computing the difference in estimated BP using the proposed techniques from ground truth values. If there is any visible anomaly in the BP reading, some optimizations can be carried out to enhance the performance of the devised system as well as improve the accuracy of the predicted BP.

Moreover, another challenge associated with NIBP implementation is the small size of datasets and the imbalanced number of volunteers, which is dominated by healthy subjects rather than diseased patients. To mitigate this matter, the incorporation of larger demographic datasets may help to identify any unusual pattern of pulse waveform, which is essential for BP analysis. Some of the inclusion criteria, including age, gender, height, weight, mental health, and history of disease, are significant prior to data acquisition to ensure the initial datasets are adequate for further analyses. Additionally, learning selection plays an important role in this part because advanced learnings, such as DL or hybridization of learnings, enable better output generation. It is worth noting that the suggested solution can be more complicated and difficult to understand for novice users, and that it can increase the computational complexity with longer training time and bigger computational memory to store the vast amount of data.

Another key challenge of NIBP technology integration is the irrelevant interchangeability of the cuff sphygmomanometer, which explains why BP values of the same subject vary when measured with different BP monitors. One of the possible reasons is that the algorithms implemented for BP interpretation are developed exclusively for every model and remain confidential to the consumers or among developers. Hence, more research efforts are currently working on seeking for the best possible alternative to measure continuous BP non-invasively and suitably for everyone through a standardized predictive-based technique using the PAT, PTT or PWV approach, and to fuse it with available learnings.

Finally, the traditional BP monitoring system configuration usually involves many wires and may cause some interference, causing less accurate BP estimation, regardless of motion disturbances. This set-up is not viable for continuous monitoring and can be further minimized into compact size with the embedding of electronics into Internet-of-Things platforms. This explains the abundance of research works exploring the aspects of material selection and structural engineering of wearable sensors to integrate this powerful device into real-time human health monitoring and improving the life quality of mankind.

## 5. Conclusions

Rapid technological advances have led to the development of optical, electrical, pressure, and ultrasonic sensors in searching for the current (gold) standard NIBP replacement that can capture data continuously in real time and can be accessed remotely. These sensors portray promising features, such as improved prediction accuracy, continuous data measurement, and mercury-free operation. Successful BP estimation using algorithms is a step forward toward the emergence of smart human health monitoring to keep up with the future trend through the development of accurate, cost-effective, comfortable, and clinically-approved continuous NIBP monitoring systems. In conclusion, the realization of continuous NIBP technology is possible considering the existing shortcomings are taken into account and resolved systematically for the enhanced future of human health monitoring and seamless data transmission.

## Figures and Tables

**Figure 1 sensors-22-06195-f001:**
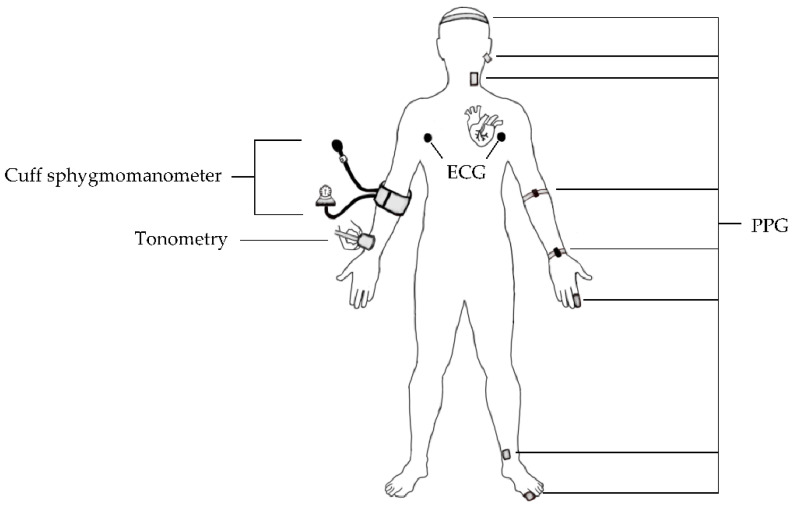
Overview of the current NIBP sensors.

**Figure 2 sensors-22-06195-f002:**
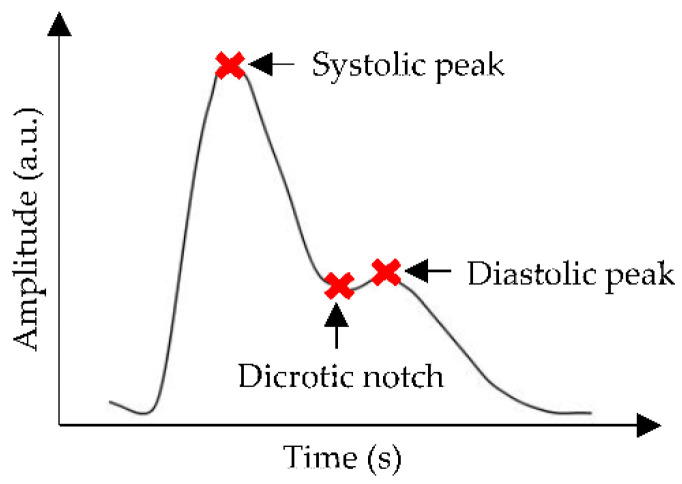
A common representation of PPG waveform, where the red cross marks show systolic peak, dicrotic notch, and diastolic peak points.

**Figure 3 sensors-22-06195-f003:**
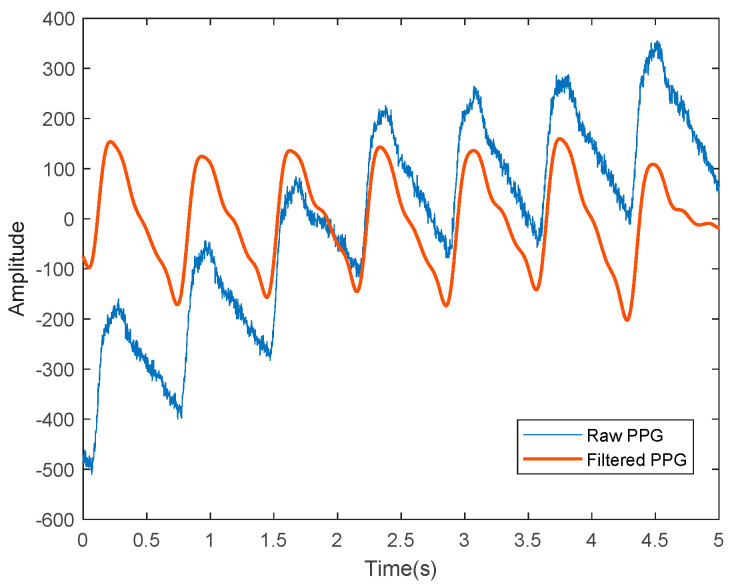
A representation of the PPG signals, where the blue line represents the raw PPG signal before denoising and the red line shows the filtered PPG signals that are ready for further analyses.

**Figure 4 sensors-22-06195-f004:**
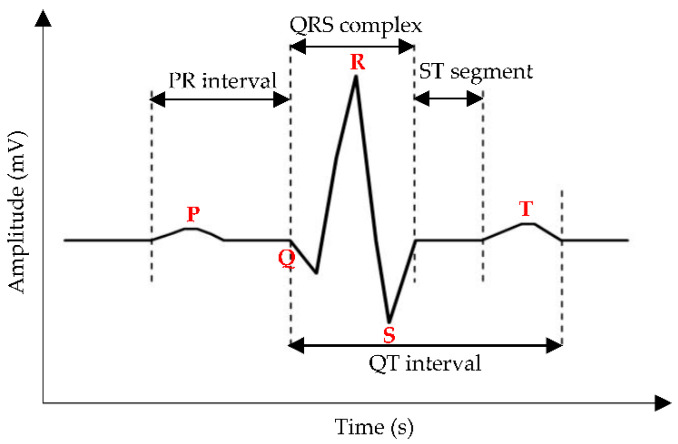
A common representation of the ECG waveform, where the red bold words indicate the P wave, the QRS complex, and the T wave. The measured distance between (1) onset of the P wave to the onset of the QRS complex, (2) offset of the S wave to the onset of the T wave, and (3) onset of the Q wave to the offset of the T wave, are labelled as PR interval, ST segment, and QT interval, respectively.

**Figure 5 sensors-22-06195-f005:**
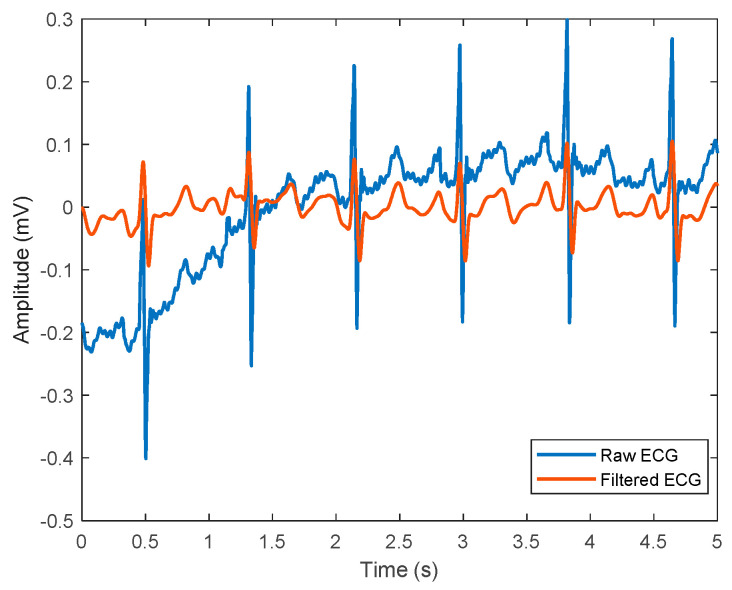
A representation of ECG signals, where the blue line represents raw ECG signal before denoising, and the red line shows filtered ECG signals that are ready for further analyses.

**Figure 6 sensors-22-06195-f006:**
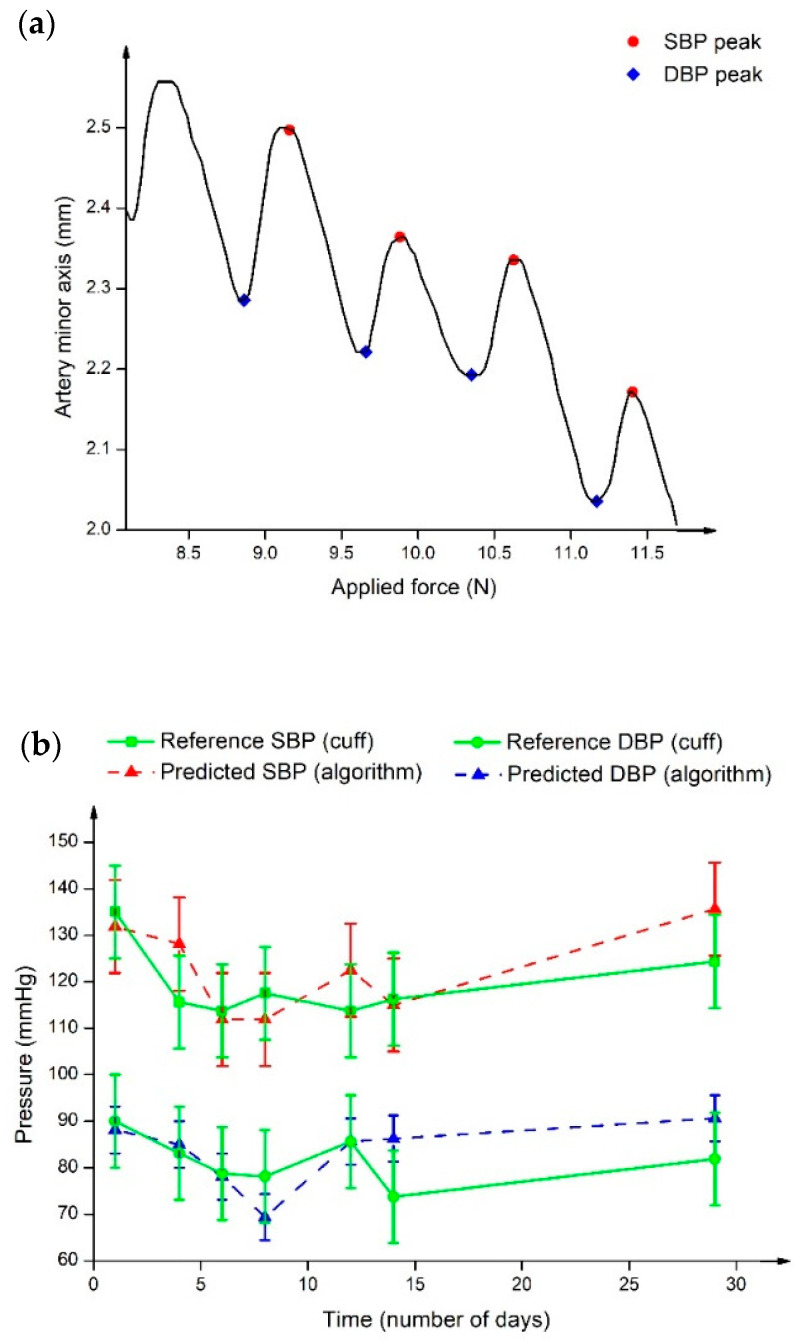
(**a**) Arterial tissue displacements against applied force graph. The arterial size changes showing internal BP pulse upon application of external pressure. Different marker colours define SBP (red) and DBP (blue) locations at the peak and trough of pressure waveform. (**b**) BP sweep of hypertensive subjects in one month, where the green line represents the reference BP from the cuff sphygmomanometer, and the red and blue dashed lines show the predicted SBP and DBP, respectively.

**Figure 7 sensors-22-06195-f007:**
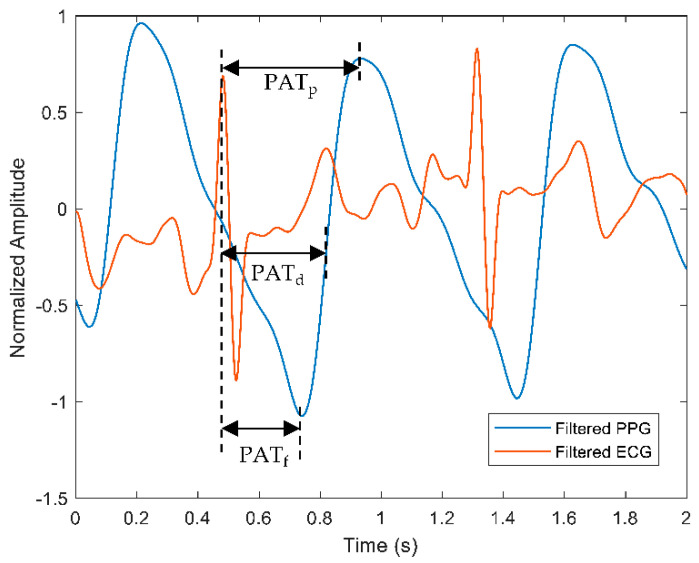
A representation of PAT calculation measured from the R-peak of the ECG signal to specific points of the PPG signal, such as peak of the PPG (${\mathrm{PAT}}_{p}$), midpoint of the PPG (${\mathrm{PAT}}_{d}$), and foot of the PPG (${\mathrm{PAT}}_{f}$). Red and blue waveforms represent filtered ECG and PPG, respectively.

**Figure 8 sensors-22-06195-f008:**
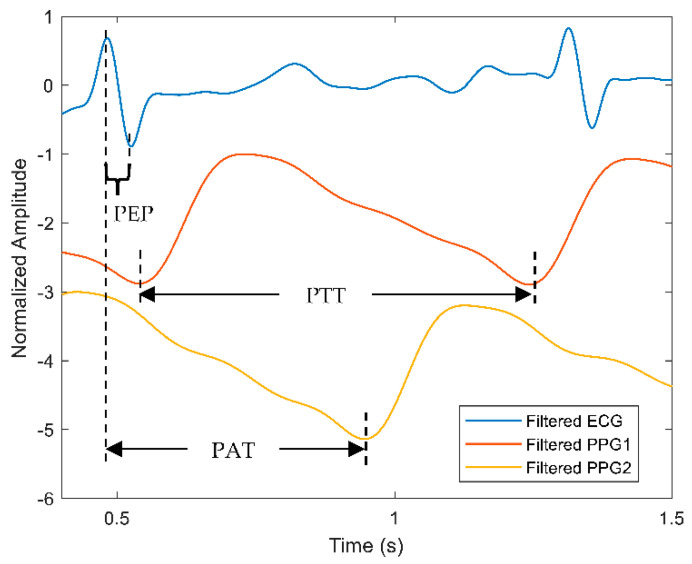
A comparison of PEP, PAT and PTT calculation measured from the ECG and PPG signals. Blue line represents the ECG signal while red and orange lines describe the filtered PPG signals from two sources: the first and second PPG signals are labelled as PPG1 and PPG2, respectively.

**Figure 9 sensors-22-06195-f009:**
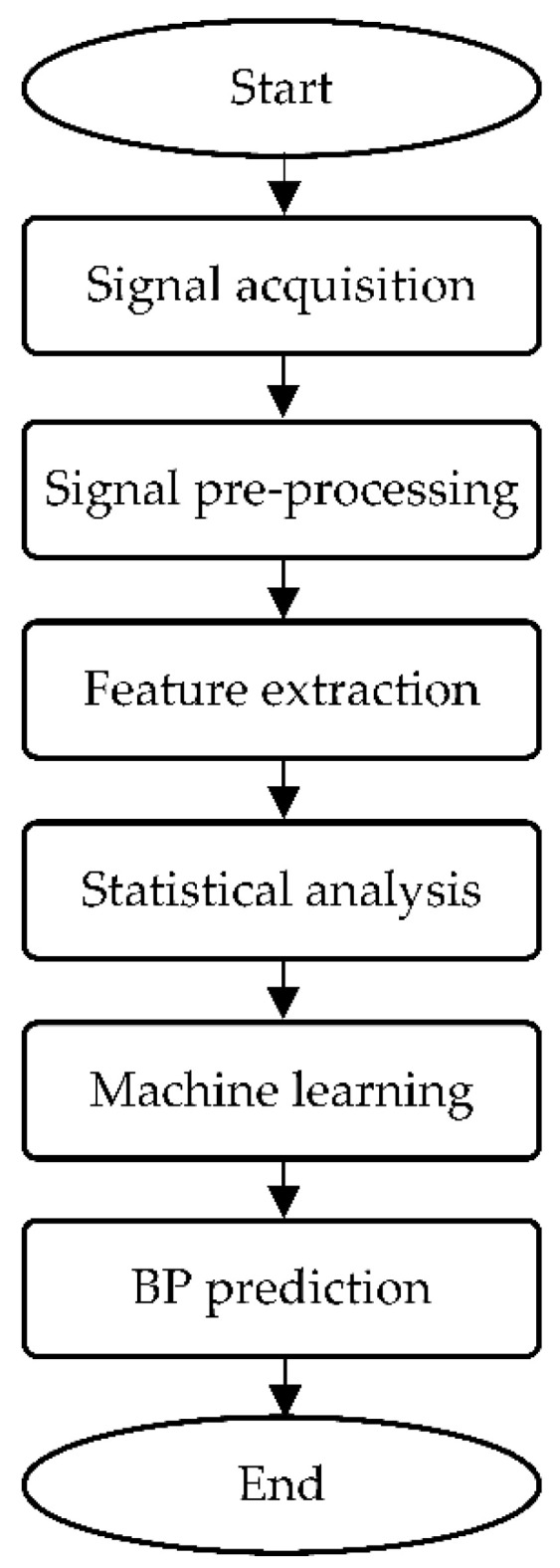
Flowchart of signal processing analysis for BP prediction.

**Table 1 sensors-22-06195-t001:** Validation of algorithms based on AAMI standard.

* Ref.	Source	Model	Feature	Subjects	BP	MAE (mmHg)	SD (mmHg)	Grade
AAMI				≥85	SBP/DBP/MAP	≤5	≤8	Pass
[39]	PPG-PPG	PWV, 14 regressors	PPG	26 (22 male, 4 female)	SBP DBP MAP	2.117 2.935 -	0.257 0.721 -	Low dataset
[100]	Single PPG	RT, MLR, SVM	PPG	Queensland 32	SBP DBP MAP	−0.1 −0.6 -	6.5 5.2 -	Low dataset
[103]	PPG, ABP	KNN, ANN, SVM	PRV	MIMIC II 500	SBP	4.74	2.33	Pass
					DBP	1.78	0.14	Pass
					MAP	2.55	0.78	Pass
[104]	Single PPG, ABP	MLR, Bi-RNN, Attention mechanism		MIMIC II 947	SBP	−0.48	9.15	Pass
			52 PPG		DBP MAP	−0.49 -	5.10 -	Pass NA
[109]	PPG, ECG, ABP			MIMIC II 379	SBP	−1.23	12.80	Fail
		1D CNN	Raw		DBP MAP	0.13 -	7.54 -	Pass *** NA
[109] ** Cal.	PPG, ECG, ABP	1D CNN	Raw	MIMIC II 379	SBP	−1.29	7.58	Pass
					DBP	−0.48	5.08	Pass
					MAP	-	-	NA
	PPG	U-Net, MultiResUNet		MIMIC II 942	SBP	−1.582	10.688	Fail
[110]			PPG		DBP	1.619	6.859	Pass
					MAP	0.631	4.962	Pass
				MIMIC 6972	SBP	4.41	6.11	Pass
[113]	PPG, ECG	CNN-LSTM	Raw		DBP	2.91	4.23	Pass
					MAP	2.77	3.88	Pass
	PPG, ECG, ABP	LR, DT, SVM, AdaBoost, RF	PPG, PAT, HR	MIMIC II 942	SBP	−0.06	9.88	Fail
[115]					DBP	0.36	5.70	Pass
					MAP	0.16	5.25	Pass
[115] Cal.	PPG, ECG, ABP	LR, DT, SVM, AdaBoost, RF	PPG, PAT, HR	MIMIC II 57	SBP DBP MAP	5.45 3.52 -	8.21 4.31 -	Low dataset

* Ref. = Reference. ** Cal. = Calibration. *** NA = Not available.

**Table 2 sensors-22-06195-t002:** Validation of algorithms based on BHS protocol.

* Ref.	Source	Model	Feature	Subjects	BP	Absolute Difference			Grade
						≤5	≤10	≤15	
BHS					SBP/DBP/MAP	60%	85%	95%	A
						50%	75%	90%	B
						40%	65%	85%	C
[109]	PPG, ECG, ABP			MIMIC II 379	SBP	40.6%	67.5%	80.2%	D
		1D CNN	Raw		DBP MAP	64.1% 62.0%	87.1% 87.1%	95.0% 95.8%	A A
[109] ** Cal.	PPG, ECG, ABP	1D CNN	Raw	MIMIC II 379	SBP	59.6%	87.3%	93.7%	B
					DBP MAP	79.2% 79.7%	95.3% 96.0%	97.9% 99.2%	A A
[110]	PPG	U-Net, MultiResUNet	PPG	MIMIC II 942	SBP	70.8%	85.3%	90.9%	B
					DBP	82.8%	92.2%	95.7%	A
					MAP	87.4%	95.2%	97.7%	A
[113]	PPG, ECG	CNN-LSTM	Raw	MIMIC 6972	SBP	67.66%	89.82%	96.82%	**A**
					DBP	82.79%	96.12%	99.09%	**A**
					MAP	84.21%	97.38%	99.58%	**A**
[115]	PPG, ECG, ABP	LR, DT, SVM, AdaBoost, RF	PPG, PAT, HR	MIMIC II 942	SBP	34.1%	56.5%	72.7%	D
					DBP	62.7%	87.1%	95.7%	A
					MAP	54.2%	81.8%	93.1%	B

* Ref. = Reference. ** Cal. = Calibration.

## Data Availability

Not applicable.

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
