# Peer review of "Recent Advances in Non-Invasive Blood Pressure Monitoring and Prediction Using a Machine Learning Approach"

_sensors, 2022, doi:10.3390/s22166195_

Round 1

Reviewer 1 Report

In this review article, the authors summarize past works on developing sensors and computational models (machine learning or deep learning) for non-invasive blood pressure monitoring. The manuscript is well written with a clear organization. I have a few comments that I wish the authors would consider.

Major comments

1.     The authors did a great job in summarizing previous studies on various types of sensors and model development. What would also be interesting to readers is knowing their view on how the differences among these options (different sensors or different models) matter to future studies / implementation of these technologies in real life. Each technique has its unique pros and cons for different scenarios (application, sample quality, sample size etc.). This paper would be more insightful if the authors include their own opinion on this perspective for Section 2 and 3.

2.     Related to the previous point, I suggest adding some discussions about validation according to different standards in Sections 3.4 and 3.5. Is there a particular model that outperforms the others? Why some fail and some pass? How does ML compare to DL? Some discussions here can shed light on future studies.

3.     The figures used in this manuscript are (presumably) copied from previous studies. The authors should contact authors of those studies for a permission to reuse the figures.

4.     Related to the last point, the resolution of figures is not good enough for publication. Some annotations are not quite intelligible. The authors should either reproduce these figures or reach out to authors of these previous papers for copies with higher resolution.

5.     Table 1 and 2 are good in general, but it would look better if the authors can make the columns more separated. Also, “Cal.” should be defined in a footnote.

Minor comment:

1.     Line 65: “… the current (gold) standard…”

Reviewer 2 Report

Paper is very important for health informatics; it needs some revisions before publication-

(1)    What are the motivations behind the selection of machine learning approach? Author must add in the paper.

(2)    Abstract must have all informations viz. problem, motivation, solution, proposed techniques and fruitful results in terms of performance evaluating parameters such as confusion matrix parameters.

(3)    Add section 2 as related works. It should consists of methodology, and demerits of previous techniques.

(4)    Resolutions of all figures must be high.

(5)    Why two butterworth outputs are only shown? Is third order was appropriate? Author must explain in the paper.

(6)    What type of software or tool was utilised?

(7)    What are the important parameters in the proposed research work?

(8)     Abstract should be enhanced by adding some Figures of Merits (FoMs) results like sensitivity, accuracy, etc.

(9)    Add motivations behind the propose research work.

(10)  More explanation on presented results is expected.

(11)  The sequence of the paper should be- (i) introduction, (ii) Related works, (iii) materials and methods, (iv) results, (v) discussion, (vi) conclusion, and (vii) future work

(12)  All tables and figures should be explained clearly.

(13)  Author should add future scope of the paper.

(14)  Add important flow charts of the work.

(15)  What is the computational complexity of the proposed technique? Author must compare with the existing techniques.

(16)  Author must check the English of the paper in the presence of native English speaker.

(17)  Compare the proposed research work with the existing research works.

(18)  The suggested papers are related papers. These must be cited in the paper-

-Wireless medical sensor network for blood pressure monitoring based on machine learning for real-time data classification

-BP Signal Analysis Using Emerging Techniques and its Validation Using ECG Signal

-Advanced artificial intelligence in heart rate and blood pressure monitoring for stress management

-Novel framework based on deep learning and cloud analytics for smart patient monitoring and recommendation (SPMR)

-Internet of Health Things (IoHT) for personalized health care using integrated edge-fog-cloud network

-Wireless IoT and Cyber-Physical System for Health Monitoring Using Honey Badger Optimized Least-Squares Support-Vector Machine

-IoT for Health Monitoring System Based on Machine Learning Algorithm

-Predictive Analysis and Prognostic Approach of Diabetes Prediction with Machine Learning Techniques

-Levenberg–Marquardt-Based Non-Invasive Blood Glucose Measurement System

-Coronary Artery Heart Disease Prediction: A Comparative Study of Computational Intelligence Techniques

-An Automated Algorithm to Extract Time Plane Features from the PPG Signal and its Derivatives for Personal Health Monitoring Application

-Impulsive Behavior Detection System Using Machine Learning and IoT

Reviewer 3 Report

This article is a review about non-invasive blood pressure measurement methods. Its target is a bit unclear, but they seem to favor ambulatory (24h) measurements only.

The scope and purpose is unclear and should be improved. It is now not a well-explained review of the field, but also not a well-structured survey with clearly defined methodology, etc. It is something in between.

I am missing related reviews/surveys. This is not the first one, so a motivation why we need another one. Here are an example of a related review article from Sensors:

Athaya, T.; Choi, S. A. Review of Noninvasive Methodologies to Estimate the Blood Pressure Waveform. Sensors 2022, 22, 3953. https://doi.org/10.3390/s22103953

A better methodology, which is also explained in the article would strengthen the article significantly. Add a clear definition of what blood pressure is sought. What are the requirements? Accuracy? Ambulatory? Comfort? Only for hypertensive patients? Hypotension as well (e.g., during ICU / anesthetics)? Other clinical needs? Cohort: Neonates/infants/adults? Sports? Other criteria? Then complement with a structured search and selection methodology.

I am missing the video-based approaches. Maybe this is due to non-expressed criteria (i.e., out of scope). But then mention that in the article. Here is one review article about video-based approaches:

Steinman J, Barszczyk A, Sun HS, Lee K, Feng ZP. Smartphones and Video Cameras: Future Methods for Blood Pressure Measurement. Front Digit Health. 2021 Nov 12;3:770096. doi: 10.3389/fdgth.2021.770096.

Finally, the following very new publication should be in your next version:

https://news.utexas.edu/2022/06/20/blood-pressure-e-tattoo-promises-continuous-mobile-monitoring/

Some Detailed Comments:

Fig. 1. Is it from [39]? Do you have copyright permission?

Line 146 is difficult to understand.

Whole Section 3.1 is difficult to follow. Increase level of explanation.

Eq. 7 is not the definition of MAP, but an approximation. MAP is defined as the arithmetic mean of the blood pressure curve during one heart cycle.

Table 1 and 2: What is the Author column? Looks like references, but [170-174] do not exist.

Reviewer 4 Report

• Which methods are used to model relationships between variables?  • The descriptions and other descriptive values/data should be defined on the tables and shapes. • Are the data subjected to pre-processing?  • How were extreme/outlier values in the data determined and resolved? 

• What approaches were used to test the validity of the models? • Which metrics were used in the performance evaluation of the estimates of models/algorithms?  • How were the predictive models selected in this study?

• How was the most suitable cut-off point determined using the receiver operator characteristic (ROC) curve analysis?

• Which method(s) was/were used to optimize the hyperparameters of models/algorithms?

Round 2

Reviewer 2 Report

Accepted in current form

Reviewer 4 Report

Acceptable